# A Comparative Study to Investigate the Effects of Bisoprolol in Patients with Chronic Heart Failure and Hypertension When Switched from Tablets to Transdermal Patches

**DOI:** 10.3390/jpm13050785

**Published:** 2023-05-01

**Authors:** Akira Sezai, Hisakuni Sekino, Makoto Taoka, Shunji Osaka, Masashi Tanaka

**Affiliations:** Department of Cardiovascular Surgery, Nihon University School of Medicine, Sekino Hospital, 30-1 Oyaguchi-kamimachi, Itabashi-ku, Tokyo 173-8610, Japan; sekinoh@sekino-hospital.com (H.S.); taoka.makoto@nihon-u.ac.jp (M.T.); osaka.shunji@nihon-u.ac.jp (S.O.);

**Keywords:** bisoprolol, beta-blocker, percutaneous, transdermal patch, heart failure

## Abstract

Background: Oral beta-blockers are effective for heart failure and hypertension. Here, we conducted a prospective study to investigate the efficacy of the beta-blocker bisoprolol in patients switching from the oral tablet to the transdermal patch. Methods: We studied 50 outpatients receiving oral bisoprolol for chronic heart failure and hypertension. After patients switched treatments, we measured heart rate (HR) over 24 h by Holter echocardiography as the primary endpoint. Secondary endpoints were (1) HR at 00:00, 06:00, 12:00, and 18:00, (2) the total number of premature atrial contractions (PACs) over 24 h and the incidence rate per time segment, and the total number of premature ventricular contractions (PVCs) over 24 h and the incidence rate per time segment, (3) blood pressure, (4) atrial natriuretic peptide and B-type natriuretic peptide, and (5) echocardiography. Results: Minimum, maximum, mean, and total HR over 24 h was not significantly different between the two groups. Mean and maximum HR at 06:00, total PACs, total PVCs, and PVCs at 00:00 to 05:59 and 06:00 to 11:59 were significantly lower in the patch group. Conclusion: Compared with oral bisoprolol, the bisoprolol transdermal patch lowers HR at 06:00 and inhibits the onset of PVCs during sleep and in the morning.

## 1. Introduction

The efficacy of oral beta-blockers in heart failure has been demonstrated in numerous large-scale trials, and they are one of the drugs needed for the treatment of heart failure [1,2]. Among beta-blockers, bisoprolol is preferred because it has the highest selectivity for the β_1_ receptor and fewer adverse reactions in patients with bronchial asthma and diabetes [3]. Transdermally administered beta-blockers containing bisoprolol were originally developed in Japan; bisoprolol was approved for the treatment of essential hypertension in 2013, and then its indications were expanded to include tachycardiac atrial fibrillation in 2019. This drug is administered once daily to achieve a stable plasma concentration. It reportedly achieves stable blood pressure lowering and heart rate lowering effects over 24 h [4]. Moreover, Matsuoka et al. reported that the 8 mg transdermal patch maintains a sustained plasma concentration of bisoprolol while lowering the peak plasma bisoprolol concentration and has a higher trough concentration than the oral 5 mg tablet; the area under the curve of plasma concentrations was similar to that of the 5 mg oral tablet [5]. Therefore, we conducted a prospective clinical study to evaluate the effects of switching patients from oral bisoprolol tablets to bisoprolol transdermal patches by measuring pulse rates and other parameters.

## 2. Methods

### 2.1. Study Protocol

This was an open-label, non-randomized trial that lasted 6 months. Study participants were outpatients who were receiving standard treatment for chronic heart failure and hypertension, had been treated with a beta-blocker for at least 6 months, and had stable disease without dose modifications. Afterward, they switched to using a bisoprolol transdermal patch (Bisono^®^ Tape, TOA EIYO Ltd. Astellas Pharma Inc., Tokyo, Japan). Chronic heart failure was defined as heart failure in patients currently on oral medications for the treatment of heart failure, such as diuretics, β-blockers, and renin–angiotensin system inhibitors. The data at the time of oral bisoprolol administration was used as the tablet group, and the data after switching from oral bisoprolol to the transdermal patch was used as the patch group.

Baseline data were obtained before switching from oral bisoprolol to the patch, and monitoring was continued for 6 months after the medication switch (Figure 1). Oral bisoprolol tablets were taken orally after breakfast, and a bisoprolol transdermal patch was attached to the chest or upper arm after breakfast. The comparative test doses of the bisoprolol transdermal patch and oral bisoprolol were 2 mg to 1.25 mg, 4 mg to 2.5 mg, and 8 mg to 5 mg, respectively. A Holter electrocardiogram (ECG) and echocardiography were conducted before patients switched to the bisoprolol transdermal patch at 0 months and again at 6 months.

This study was conducted at Sekino Hospital, a logistical support hospital of the Nihon University Itabashi Hospital, according to the Declaration of Helsinki. The study details were explained to patients, and written informed consent was obtained. The study was approved by our institutional review board and registered with our Hospital Medical Information Network (study ID: UMIN000031538).

The primary endpoints were measurements of minimum, mean, maximum, and total heart rate over 24 h determined by using a Holter ECG before (at 0 months) and 6 months after the medication switch.

The secondary endpoints were as follows: (1) minimum, maximum, and mean heart rate at 00:00, 06:00, 12:00, and 18:00 measured by Holter ECG (measurements at 6:00 and 00:00 were conducted during sleep); (2) the total number of premature atrial contractions (PACs) over 24 h (excluding atrial fibrillation), the number of events in each time interval, and the total number of premature ventricular contractions (PVCs) over 24 h; (3) home blood pressure measurements (systolic and diastolic) in the early morning; (4) atrial natriuretic peptide (ANP) and B-type natriuretic peptide (BNP) levels measured at 0, 3, and 6 months; (5) echocardiography (0 months and 6 months after medication switch), left ventricular ejection (EF), fractional shortening (% FS), left ventricular end-diastolic dimension (LVDd), left ventricular end-systolic dimensions (LVDs), and the left ventricular mass index (LVMI).

The following adverse effects were recorded: hypotension, bradycardia, renal dysfunction (defined as an increase in serum Cr levels by ≥50%), hepatic dysfunction (defined as an increase in AST/ALT by ≥50%), skin reactions, and allergic reactions. The management of adverse reactions, which included the discontinuation of the bisoprolol transdermal patch, was decided by the attending physician.

### 2.2. Statistical Analysis

Observed values were expressed as medians and 25th and 75th percentiles. ANP and BNP were analyzed by Friedman’s test, and the other variables by the Wilcoxon signed rank test. A *p*-value of less than 0.05 was considered statistically significant.

## 3. Results

Fifty patients were enrolled in this trial, and their baseline characteristics are shown in Table 1. In accordance with the study protocol, patients switched from the oral tablet to the transdermal patch and remained on it for 6 months. Five patients discontinued the transdermal patch during the study; the reasons for discontinuation were the development of skin rashes in four patients (8%) and the development of bradycardia in one patient (2%). Except for the five discontinued cases, all data were analyzed.

Primary endpoints are shown in Table 2. There were no significant differences between the two groups in the minimum, maximum, mean, and total HR over 24 h.

The results of the secondary endpoints are summarized below:

(1) The hourly heart rates are shown in Table 3. For heart rates measured at 6:00, there were no differences in the minimum heart rates between the groups (*p* = 0.334). However, the mean heart rates at 06:00 were 72.9 ± 2.1 bpm and 67.7 ± 1.6 bpm in the tablet and patch groups, respectively (*p* = 0.018), and the maximum heart rates at 06:00 were 90.0 ± 2.9 bpm and 80.6 ± 2.6 in the tablet and patch groups, respectively (*p* = 0.002). Thus, the hourly heart rate was significantly lower in the patch group than in the tablet group. No group differences were observed at other time points.

(2) We determined the total PACs and PVCs that occurred over 24 h and the PACs and PVCs per time interval (Figure 2, Table 2). Both total PACs and PVCs were significantly lower in the patch group than in the tablet group (PAC, *p* = 0.015; PVC, *p* = 0.039; Figure 2). PACs in terms of the onset per hour were not significantly different between the groups, but PVCs were significantly lower at 00:00 to 05:59 and 06:00 to 11:59 in the patch group (Table 2). Even though PVCs were observed in both groups, the majority of occurrences were isolated PVCs, and ablation therapy was not needed. In this study, mexiletine hydrochloride was orally administered for PVCs, and the dose was not modified during the study period.

(3) Systolic and diastolic blood pressure were measured at home in the early morning (Table 2), and there were no significant differences in either systolic or diastolic blood pressure measurements between the two groups.

(4) ANP and BNP levels did not change significantly over time (Table 4).

(5) Echocardiographic data showed that EF, LVDd, LVDs, E/e’, and the LVMI were not significantly different between the two groups (Table 4).

## 4. Discussion

This study indicated that the patch group had significantly lower HRs in the early morning. The total number of PACs and PVCs in 24 h was significantly lower in the patch group than in the tablet group, and the number of PVCs was significantly lower during sleep and in the morning in the patch group. With regard to the onset of arrythmia, significantly fewer PACs and PVCs occurred over 24 h in the patch group than in the tablet group, and the number of PVCs were significantly lower during sleep and in the morning in the patch group. We selected 06:00 as the study time point because cardiovascular events and sudden death are more likely to occur in the morning and during sleep. The study result shows that the effects of the patch formulation of beta-blockers last longer than those of the tablet formulation. However, it is not investigated in patients who require treatment for arrhythmia or tachycardia. As such, it is necessary to study the effects of both drug formulations in patients who need to be treated for arrhythmia or tachycardia.

In a previous study, we reported that the peak blood bisoprolol concentration after 2 weeks of treatment was 2.2 ± 0.8 h in the tablet group and 7.8 ± 2.0 h in the patch group and that the half-life was 10.02 ± 1.29 h in the tablet group and 20.80 ± 4.48 h in the patch group [6]. Blood bisoprolol concentrations were higher in the tablet group in the first 6 h after administration of the medication, and thereafter, the patch group had higher blood bisoprolol concentrations from 6 to 36 h after administration of the medication. In the present study, we did not measure the blood concentration of bisoprolol; however, because the half-life of the patch formulation is longer and the blood bisoprolol concentration from 6 to 36 h after the application is higher, we presume that the efficacy of the drug persisted during sleep and through to the morning of the next day.

Until now, only two studies have evaluated the switch from tablets to patches. Momomura et al. performed a phase II study on 40 patients with chronic heart failure who switched from oral tablets to patch formulations; the study showed favorable safety and efficacy. Based on the New York Heart Association functional classifications, there were no changes in the left ventricular functions observed by cardiac ultrasound imaging. Moreover, there were no pulse rate changes after switching to the patch. However, systolic blood pressure was significantly lower at weeks 8 and 16 after the medication switch [4]. Sairaku et al. reported that 30 patients with hypertension had their medication changed from the orally administered tablet to the patch formulation, and no differences were observed in the 24 h time domain or frequency domain of the heart rate variability (HRV) measurements. However, switching to the patch significantly altered the time-course curves of the hourly HRV measurements, which included the mean normal-to-normal (NN) interval, the standard deviation of the NN index, the high-frequency component, and the low-frequency component. Even though an equivalent dose of bisoprolol was given, the authors concluded that the autonomic modulation pattern might vary depending on whether the patient received the bisoprolol transdermally or orally [6].

To date, only two randomized studies on patients allocated to either tablet or patch versions of bisoprolol have been reported. Yamashita et al. conducted a comparative study of oral versus patch formulation in 220 patients with persistent and chronic atrial fibrillation (BBISONO-AF study). No group differences were observed in either the resting or mean HRs over 24 h measured by 12-lead ECG. They also reported that a 2.5 mg tablet is comparable to a 4 mg patch, and a 5 mg tablet is comparable to an 8 mg patch. Moreover, they found that the HR-lowering effect of the transdermal patch is most effective in the early morning when the HR is rising or when a high HR condition causes sympathetic input. Therefore, the patch was more effective in patients with sympathetic nervous tension [7]. Matsuoka et al. demonstrated that the patch had a more stable plasma concentration time profile than the tablets, and the morning heart rate was dose-dependent and significantly lower in the patch group than the tablet group [5].

Neither of these studies reported data on PVCs or PACs. However, Shinohara et al. investigated the efficacy of the bisoprolol patch by measuring the frequency of PVCs in 44 patients without structural heart disease. They reported a consistent decrease in PVC over 24 h and wrote that PVCs are triggered by sympathetic activation and that the patch formulation of bisoprolol has a longer half-life and exerts sustained effects in reducing PVCs over 24 h [8].

Concerning adverse effects, skin rash was observed in four patients in the present study. Initially, the patch was manufactured with a rubber adhesive, but an acrylic adhesive started to be used in January 2019. In this study, 23 patients used rubber adhesive patches only, 16 patients used acrylic adhesive patches only, and 11 patients used rubber adhesive patches at first and then switched to acrylic adhesive patches. Skin rash was observed in three patients (8.85%) with a rubber adhesive patch and one patient (3.7%) with an acrylic adhesive patch; the difference was not significant (*p* = 0.623). However, the question of whether the type of adhesive affects the rate of skin rash may need further investigation.

In this study, the test drug, the bisoprolol transdermal patch, was administered for 6 months, which was the longest observation period reported so far. This study demonstrated stable effects over 24 h and clearly suggested that there was parasympathetic nerve inhibition throughout the day. Although there were no significant differences in the LVMI at 6 months, a decreasing trend was observed. Therefore, a longer treatment period may result in improvement and inhibition of cardiac enlargement and prognosis. Thus, an additional study with a longer observational period is necessary to better understand these results.

## 5. Limitations

This study has several limitations. Although it was a prospective study, it was performed at a single center with a limited number of patients. Additionally, we only investigated the effects in patients who switched from the orally administered tablet to the transdermal patch and did not evaluate patients who switched from the patch to oral tablets. In addition, the study did not measure blood concentrations of bisoprolol, and obtaining such data would help to determine the differences between the patch and oral formulations. A 2.5 mg tablet is considered to correspond to a 4 mg patch, and a 5 mg tablet to an 8 mg patch. However, because we did not measure blood concentrations during the study period, we do not know whether there is a correlation between the doses of the two formulations. Furthermore, it is difficult to conclude whether the observed effects were due to the dose or differences in the dose because of the small number of patients. Therefore, a future study with a more robust design is required.

## 6. Conclusions

Compared with oral bisoprolol, the bisoprolol transdermal patch lowers HR at 06:00 and inhibits arrythmia during sleep and in the morning.

## Figures and Tables

**Figure 1 jpm-13-00785-f001:**
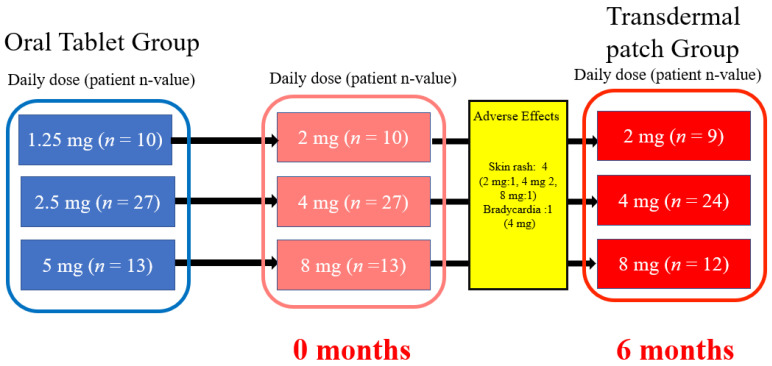
Study flowchart.

**Figure 2 jpm-13-00785-f002:**
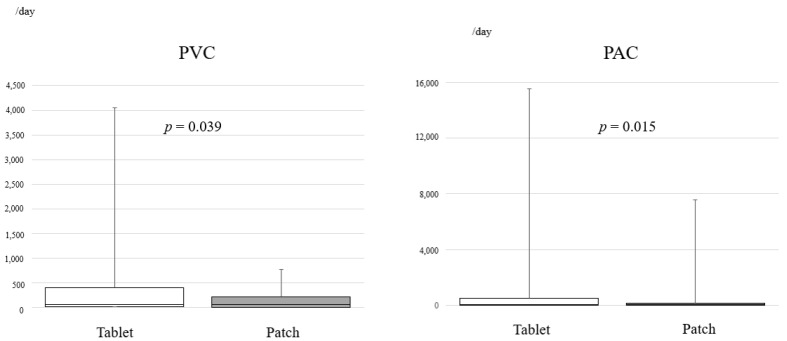
Bar graph showing the total number of premature ventricular contractions and premature atrial contractions over 24 h in patients from the oral bisoprolol tablet group and the bisoprolol transdermal patch group. PVCs, premature ventricular contractions; PACs, premature atrial contractions.

**Table 1 jpm-13-00785-t001:** Patient characteristics.

Total number of patients	50
Age (years)	75.1 ± 9.5
Sex: male, female	28, 22
Basic disease	n (%)
Ischemic heart disease	13 (26%)
Valvular disease	22 (44%)
Hypertensive heart disease	13 (26%)
Other	2 (4%)
Classification of heart failure	n (%)
HFrEF	6 (12%)
HFmrEF	5 (10%)
HFpEF	33 (66%)
HFpEF improved	6 (12%)
Risk factors	n (%)
Type 2 diabetes	18 (36%)
Dyslipidemia	41 (82%)
Hyperuricemia	25 (50%)
Atrial fibrillation	17 (34%)
Obesity	9 (18%)
Medications	n (%)
Oral bisoprolol	
1.25 mg	10 (20%)
2.5 mg	27 (54%)
5.0 mg	13 (26%)
Calcium antagonist	17 (34%)
Angiotensin II receptor blocker	18 (36%)
ACE inhibitor	4 (8%)
Aldosterone blocker	22 (44%)
α-blocker	5 (10%)
Diuretics	17 (34%)
Oral hypoglycemic agent	18 (36%)
Statin	42 (84%)
Ezetimibe	12 (24%)
Xanthanide oxidase antagonist	25 (50%)
Antiarrhythmic drugs	n (%)
Bepridil	4 (8%)
Disopyramide	1 (2%)
Mexiletine	10 (20%)
Pilsicanide	4 (8%)
Verapamil	2 (4%)

ACE, angiotensin-converting enzyme; HFmrEF, heart failure with mid-range ejection fraction; HFpEFm, heart failure with preserved ejection fraction; HFrEF, heart failure with reduced ejection fraction.

**Table 2 jpm-13-00785-t002:** Blood pressure, heart rate, premature ventricular contractions, and premature atrial contractions over 24 h before and after switching from oral bisoprolol tablets to the bisoprolol transdermal patch.

	Tablet	Patch	*p* Value
Systolic blood pressure (mm Hg)	127 (117, 137.5)	127.5 (116.5, 141.3)	0.674
Diastolic blood pressure (mm Hg)	71.5 (65.8, 81.3)	73.5 (66, 81)	0.977
Heart rate			
Minimum (bpm)	56 (51, 60.5)	54 (51, 61)	0.63
Mean (bpm)	73 (68.5, 80.5)	72 (66.5, 77)	0.227
Maximum (bpm)	115 (102.5, 126)	112 (97.5, 126)	0.359
Total (n)	100,865 (90,907, 109,855)	96,953 (90,365, 106,020)	0.467
PVC time segment (h)			
0–5	8 (1, 87)	4 (0, 24.5)	0.005
6–11	13 (3, 110)	8 (5, 32.5)	0.009
12–17	10 (3, 85)	13 (1, 58.5)	0.359
18–24	15 (2, 101)	15 (1, 42.5)	0.085
PAC time segment (h)			
0–5	10 (0, 46.5)	5 (0, 23.5)	0.07
6–11	8 (0, 56)	12 (0, 26)	0.062
12–17	13 (0, 53)	10 (0, 23.5)	0.269
18–24	11 (0, 28)	10 (0, 28.5)	0.33

PACs, premature atrial contractions; PVCs, premature ventricular contractions.

**Table 3 jpm-13-00785-t003:** Minimum, mean, and maximum hourly heart rate (bpm) for six hourly blocks at 0:00, 6:00, 12:00, and 18:00.

	Tablet	Patch	*p*-Value
6:00			
Minimum	60 (55, 70.5)	60 (54.5, 68)	0.239
Mean	71 (61, 81.5)	65 (61, 76)	0.018
Maximum	85 (75, 101.5)	79 (73.5, 93)	0.003
12:00			
Minimum	66 (60.5, 73)	65 (58, 74,5)	0.615
Mean	75 (70, 87.5)	74 (68, 81.5)	0.448
Maximum	92 (83, 108.5)	89 (80.5, 105)	0.368
18:00			
Minimum	68 (61.5, 73)	66 (60.5, 72.5)	0.325
Mean	77 (67, 84.5)	72 (68, 82.5)	0.114
Maximum	91 (85.5, 101.5)	87 (77.5, 100)	0.082
0:00			
Minimum	62 (55.5, 68)	60 (55, 67.5)	0.529
Mean	67 (61, 73)	65 (58.5, 71)	0.194
Maximum	78 (69, 86.5)	76 (70, 82)	0.403

**Table 4 jpm-13-00785-t004:** Patient echocardiographic measurements of atrial natriuretic peptide and B-type natriuretic peptide levels before and after switching from oral bisoprolol tablets to bisoprolol transdermal patch.

	0 Months (Tablet)	3 Months (Patch)	6 Months (Patch)	* p * -Value
ANP levels (pg/mL)	73.6 (46.9, 118.8)	65.6 (50.3, 96.1)	78.8 (45.2, 121.8)	0.341
BNP levels (pg/mL)	89.8 (41.2, 174.5)	83.4 (39.7, 174.7)	87.1 (47.9, 186.4)	0.428
Echocardiography	0 months (Tablet)		6 months (Patch)	*p*-value
LVDd (mm)	45 (43, 51)		45 (41, 52)	0.861
LVDs (mm)	29.5 (27, 34.3)		29 (27, 34,3)	0.638
Ejection fraction (%)	64.5 (59.3, 68.3)		63.7 (59.9, 67.4)	0.400
E/e′	10,7 (7.9, 17.1)		11 (8.1, 17.3)	0.369
LVMI (g/m^2^)	158.8 (121.5, 181.7)		141.1 (120.8, 164.9)	0.064

ANP: atrial natriuretic peptide, BNP: B-type natriuretic peptide, LVDd: left ventricular end-diastolic diameter, LVDs: left ventricular end-systolic diameter, E/e′: the ratio of early diastolic mitral inflow to mitral annular tissue velocities, LVMI: left ventricular mass index.

## Data Availability

In principle, the data used in this study cannot be shared.

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
