# Peer review of "A Comparative Study to Investigate the Effects of Bisoprolol in Patients with Chronic Heart Failure and Hypertension When Switched from Tablets to Transdermal Patches"

_jpm, 2023, doi:10.3390/jpm13050785_

Round 1

Reviewer 1 Report

Unfortunately,

In many respects this manuscript is similar to the one already published. 

https://www.jstage.jst.go.jp/article/circj/advpub/0/advpub_CJ-17-0532/_article/-char/en
I am not sure that it brings any news

Author Response

Dear Reviewer 1,

Thank you very much for your detailed review of our manuscript and the instructions for revisions. The manuscript has been reviewed by a native English speaker.

Our article is now ready for re-submission with the revisions detailed below and we consider that our report now has a stronger impact.

We would appreciate it if you could assess the revised manuscript.

Unfortunately, in many respects this manuscript is similar to the one already published. 
https://www.jstage.jst.go.jp/article/circj/advpub/0/advpub_CJ-17-0532/_article/-char/en
I am not sure that it brings any news.

→Thank you for your detailed review of our manuscript and the instructions for revisions. The article that you cited is actually included in the Reference list of the manuscript. The current study is different in that it analyzed Holter ECG data and intensively investigated the onset of arrhythmias and changes in heart rate throughout the day and night. As such, we believe that it presents novel findings and would appreciate it if you could review the revised version of the manuscript. 

Reviewer 2 Report

Review of manuscript jpm-2328374

Dear authors,

I have read with interest your research paper entitled:“A comparative study to investigate the effects of bisoprolol in 2 patients with chronic heart failure and hypertension when three switched from tablets to transdermal patches“. It is definitely an interesting topic to study and i congratulate you on the results. You are not wrong in choosing a clinical outcome; I would have chosen a pharmacological in the first place (plasma beta-blocker levels in both groups) and the clinical as a secondary outcome. Even if I overcome these design downsides, several major issues must be addressed. A point-by-point answers are required addressing all the bellow stated issues. Also extensive english correction is necessary.

Abstract:

-       Already the first sentence makes no sense, either :“are effective or effectively treat Heart failure and hypertension..“

-       Number of subjects participating in the study  should be stated in the abstract 

Statistical analysis: 

-       To utilize a parametric test in such a small study group is highly questionable. I suggest you recalculate all the effects with non-parametric variants of the tests, i.e. in the case you use Student’s T-test you will use Wilcoxon signed rank test. 

-       It is lso not stated, wether you aim for one- or two-tailed alpha, this should be added to the statistical analysis paragraph

-       Normality testing (i.e. Kolmogorov-Smirnov normality test) of all the continuous variables should be provided here in the stistical analysis and then in the results -i assume in this small group the data will be non-normally distributed

Results:

-       Line 97: you state that there are no significant differences in the minimum, maximum, mean and total HR over 24h in the contradiction to the line 102, where you state: „The maximum heart rates were 90.0±2.9 bpm and 80.6±2.6 102 for the Tablet and Patch groups, respectively (p=0.002)

-       With the hourly mean it should also be corrected, so over 24 hours there is no change but at 6:AM there is i a significant difference in the mean HR between Tablet vs. patch (the other question is why you are using capital letters for tablet and patch)

-       I recommend you re-write the results section to make crystal clear where were the differences in the HR between groups and where were not

-       Line 113 the sentence does not make sense, please correct. Also what you mean by „the PVC did not require treatment“. If the patients is already using betabocker in whichever form, it is already a treatment. You meant it doe not require a ablation therapy, if so, you should have express it properly.

-       Line 119, again this sentence does not make sense. „..did not significantly differ over time“ not „…different change overtime“!

Discussion:

-       The first paragraph of the discussion section should be summarization of the your results. So this paragraph should be re-written comletely. This study indicates … not indicated

-       In the discussion section your own data nad literature data are mixed, to make it clear i recommed you to divide it into subsections: current study and previous studies

-       Line 172 again Ten 10 patients…correct

-       The discussion section is merely a list of other studies with no intention to discuss. You definitely should try to explain why there were changes in the mean and maximum HR (if the difference will still b ethere after you use correct statistics)

Conclusion:

The data definitely does not support your conclusion. You stated that over 24 hours there were no differences in the HR, than in some predifined time period you have found some differences but you did not explained why and you also did not explained why you hve chosen exactly 6 AM for the test This should al l be explained. 

Author Response

Dear Reviewer 2,

Thank you very much for your detailed review of our manuscript and the instructions for revisions. The manuscript has been reviewed by a native English speaker.

Our article is now ready for re-submission with the revisions detailed below and we consider that our report now has a stronger impact.

We would appreciate it if you could assess the revised manuscript.

I have read with interest your research paper entitled:“A comparative study to investigate the effects of bisoprolol in 2 patients with chronic heart failure and hypertension when three switched from tablets to transdermal patches“. It is definitely an interesting topic to study and i congratulate you on the results. You are not wrong in choosing a clinical outcome; I would have chosen a pharmacological in the first place (plasma beta-blocker levels in both groups) and the clinical as a secondary outcome. Even if I overcome these design downsides, several major issues must be addressed. A point-by-point answers are required addressing all the bellow stated issues. Also extensive english correction is necessary.

 →Thank you for your detailed review of our manuscript and your suggestions and comments. The revised manuscript has been reviewed by a native English speaker. I attached the certificate of the medical translation service company. We agree with you that it would have been useful if we had measured the plasma concentration of beta blockers. However, we unable to perform extensive measurements because the study was conducted as part of daily clinical practice.

Abstract:

-       Already the first sentence makes no sense, either :“are effective or effectively treat Heart failure and hypertension..“

→We apologize for this mistake and have corrected it.

-       Number of subjects participating in the study should be stated in the abstract 

→Thank you for your comment. We have added the number of participants.

Statistical analysis: 

-       To utilize a parametric test in such a small study group is highly questionable. I suggest you recalculate all the effects with non-parametric variants of the tests, i.e. in the case you use Student’s T-test you will use Wilcoxon signed rank test. 

-       It is lso not stated, wether you aim for one- or two-tailed alpha, this should be added to the statistical analysis paragraph

-       Normality testing (i.e. Kolmogorov-Smirnov normality test) of all the continuous variables should be provided here in the stistical analysis and then in the results -i assume in this small group the data will be non-normally distributed

→Thank you for your very useful comment. We re-analyzed the results with more suitable tests, as you suggested.

Results:

-       Line 97: you state that there are no significant differences in the minimum, maximum, mean and total HR over 24h in the contradiction to the line 102, where you state: „The maximum heart rates were 90.0±2.9 bpm and 80.6±2.6 102 for the Tablet and Patch groups, respectively (p=0.002)

-       With the hourly mean it should also be corrected, so over 24 hours there is no change but at 6:AM there is a significant difference in the mean HR between Tablet vs. patch (the other question is why you are using capital letters for tablet and patch)

-       I recommend you re-write the results section to make crystal clear where were the differences in the HR between groups and where were not

→We apologize for the confusion. There was no significant difference in the minimum, maximum, mean, or total HR over 24 hours. However, a significant difference was observed in the mean and maximum HR at 6:00 AM. We have added additional explanations and changed the test method. Because we believe that a table may be clearer than a figure, we now present the data from former Figure 2 in the new Table 3.

-       Line 113 the sentence does not make sense, please correct. Also what you mean by „the PVC did not require treatment“. If the patients is already using betabocker in whichever form, it is already a treatment. You meant it doe not require a ablation therapy, if so, you should have express it properly.

→Thank you for this important comment. To our knowledge, there is no experience in the use of beta-blockers as PVC therapy. Ablation therapy was not required. We now comment on this point.

-       Line 119, again this sentence does not make sense... did not significantly differ over time“ not „…different change overtime“!

→Thank you for your comment. We have corrected the sentence as follow:
ANP and BNP levels did not change significantly over time (Table 4).

Discussion:

-       The first paragraph of the discussion section should be summarization of the your results. So this paragraph should be re-written comletely. This study indicates … not indicated

→We thank you for your advice. We now describe only the study results in the first paragraph.

-       In the discussion section your own data nad literature data are mixed, to make it clear i recommed you to divide it into subsections: current study and previous studies

→Thank you for your valuable advice. We now describe our own data and past study results separately.

-       Line 172 again Ten 10 patients…correct

→We apologize for the confusion. Ten patients were receiving mexiletine hydrochloride for the treatment of PVC, and the dose was not increased or decreased in these patients during the study period. This point is described in the Results section.

-       The discussion section is merely a list of other studies with no intention to discuss. You definitely should try to explain why there were changes in the mean and maximum HR (if the difference will still b ethere after you use correct statistics)

→Thank you for your valuable comment. The study results showed that the mean and maximum heart rates at 6:00 AM were significantly lower in the patch group than in the tablet group. Furthermore, the onset of arrythmia was significantly lower during sleep and in the morning in the patch group than in the tablet group. We have added this information to the first part of the Discussion.

Conclusion:

The data definitely does not support your conclusion. You stated that over 24 hours there were no differences in the HR, than in some predifined time period you have found some differences but you did not explained why and you also did not explained why you hve chosen exactly 6 AM for the test This should al l be explained. 

→Thank you for this valuable comment. We have edited the conclusion derived from the study results and now explain in the first paragraph of the Discussion why 06:00 AM was selected as the time for the test. 

Reviewer 3 Report

In the current manuscript, Sezai and colleagues compared the use of transdermal bisoprolol patches in existing heart failure patients taking oral bisoprolol and evaluated the modal shift differences on pulse rates, blood pressures, atrial/ventricular ectopy by 24h holter ECGs, ANP/BNP levels (baseline vs. 3 vs. 6mo) and echocardiography for 6 months in an open-label, non-randomized trial in 50 patients. The baseline characteristics was assessed in patients undertaking oral bisopronolol (control), with subsequent assessment (6mo) after transiting to bisopronolol patches (dosage was given at 1.6-fold that of patients' current oral dose owing to comparable efficacy on ambulatory blood pressure monitoring (ref #5). Overall, the current study found that no deleterious effects of the transition to patch-mode administration, but significantly lower HR in the mornings and reduced ventricular ectopy burden during the night and early mornings where patients are assumed to be resting. Their findings suggest that transdermal mode of bisoprolol administration may be optimal compared to oral dosing owing to the sustained and longer-acting bio-distribution of the drug.

This reviewer has the following comments:

1) Introduction line 39 - what does "AUC (infinity)" mean? Looking at the citation (ref #5), it appears that the authors mentioned that "TY-0201 (TY) is a new transdermal β-blocker formulation containing bisoprolol, and TY 8 mg was designed to maintain a sustained concentration of bisoprolol in plasma by lower peak plasma bisoprolol concentration and higher trough concentration than BO 5 mg, thereby demonstrating an area under the curve for plasma concentration similar to that of BO 5 mg." - however this piece of evidence was not explicitly presented in the paper? Please reference to original data.

Furthermore, the same pharmacokinetics studies were not applied to other doses as presented here despite patch dosage was normalized to 1.6-fold that of oral dose in patients under active bisopronolol treatment. Please discuss, and further expand on the limitations for not assessing serum drug concentrations in the current study.

2) Methods line 47 - Your definition of chronic heart failure sounds more like an inclusion criteria rather than a definition of "chronic heart failure". For trials, please clearly identify your inclusion/exclusion criteria i.e. Chronic heart failure patients currently on oral bisopronolol treatment were recruited. This study did not exclude patients on other medications (diuretics,... etc) to treat their heart failure conditions. 

Also, it isn't clear if there was a clear definition on how long patients had to be on oral bisopronolol before they could be transited to patch mode of administration. Was the limiting factor as to how long patients needed to be on this dosage before enrolment in the trial established? 

3) Figure 1 - Patients were excluded for adverse effects of the patch. However, pg 3 line 81 also states other following adverse effects were recorded, and discontinuation of the patch was decided by the attending physician (which supposedly resulted in the withdrawal of 4 patients with skin rash and 1 patient with bradycardia). What were the proportions of adverse effects reported (if any at all) after transiting to patch that were supposedly reported (other than 4x skin rash and 1x bradycardia)?

4) Statistics - There is not enough information to evaluate suitability of statistics. Firstly, this study utilizes a paired study cohort (given that all 50 patients were prior oral bisopronolol users before transition onto dermal patches). In order to perform parametric analyses, authors should check if dataset conforms to normal distribution, and test for equal variance (Welch's correction applies to unequal variances).

What was the post-hoc analyses to test multiple comparisons in ANOVA? For student t-test, was this a one- or two-tailed, paired or unpaired t-test?

5) Were there differences in any findings between the 3 doses of bisopronolol (2 vs. 4 vs. 8mg)? I note that the dataset was probably underpowered to assess such a breakdown (which is a potential limitation). It will be curious to know if severity depended on the dosage of therapy.

6) Figure 2 - (i) Y-axis should be labeled as Heart rate (bpm) and bpm should be explained as beats per minute in the legend. (ii) Bar graphs do not do justice to individual data points. Suggest to present as box-violin plots which provides inter-subject variability information. (iii) colour scheme should be revised. It is hard to tell the error bars of the patch group against the blue shades background. 

Same for data presentation in Figure 3. 

7) Table 3 - Tablet vs patch should be highlighted at the top to indicate that Tablet group was conducted at baseline (0mo) whereas Patch was conducted over both (3 and 6mo) for clarity which phase of the treatment that these measurements were conducted. Similarly, all abbreviations including E/e' should be explained in footnote.

8) Discussion - Structure of discussion needs to be improved upon. It started off with saying patch group has <HR and <PVC which are findings from this current study. This is followed by results of blood concentration levels of bisoprolol (which was not conducted in the current study)...  cited reference  (ref #6) utilized 2.5mg tablet vs. 4mg patch. How does this actually translate to the other dosages engaged in the current study?

Line 186-188: Sentence structure needs revision. Appears to suggest that the mechanism of bisoprolol (beta-blocker) increased cardiac contractions and heart rates, which is a contradiction.

Line 193-194: Revise sentence - I don't understand the relevance of "Ten" 10 patients receiving mexiletine hydochloride and the implication of the subsequent sentence of reducing anti-arrhythmic drug dose and decrease in sudden death in context of your discussion?

9) Discussion line 198 to 201 - Surprisingly, new results are presented in the discussion. This result is weird... and sentence structure should be amended to better communicate the findings. 

39 patients used rubber adhesives vs 26 patients used acrylic adhesives = 61 patients. Obviously, study recruitment was 50, so some alternated in between (n=11). However subsequent sentence state 11 patients used only acrylic adhesive, but 15 switched from rubber to acrylic. What about patients that switched from acrylic to rubber? and why am i having to calculate the proportions when this should be clearly presented i.e. how many used rubber adhesive throughout/ acrylic adhesive throughout/ alternated from rubber>acrylic adhesive/ acrylic>rubber adhesive? was there any control/regulation as how these were decided i.e. were patients randomly given a mix of the patches (acrylic/rubber) or did patients have side effects and therefore required to change patches?

Minor comments:

1) Abstract line 16 - consider changing text to 06:00, 12:00, 18:00, and 00:00 hours or AM/PM instead of "o'clock".

2) Table 1. All abbreviations should equally be explained in the footnote, rather than just expanding "ACE".

Author Response

Dear Reviewer 3 

Thank you very much for your detailed review of our manuscript and the instructions for revisions. The manuscript has been reviewed by a native English speaker.

Our article is now ready for re-submission with the revisions detailed below and we consider that our report now has a stronger impact.

We would appreciate it if you could assess the revised manuscript.

  • Introduction line 39 - what does "AUC (infinity)" mean? Looking at the citation (ref #5), it appears that the authors mentioned that "TY-0201 (TY) is a new transdermal β-blocker formulation containing bisoprolol, and TY 8 mg was designed to maintain a sustained concentration of bisoprolol in plasma by lower peak plasma bisoprolol concentration and higher trough concentration than BO 5 mg, thereby demonstrating an area under the curve for plasma concentration similar to that of BO 5 mg." - however this piece of evidence was not explicitly presented in the paper? Please reference to original data.
  • Furthermore, the same pharmacokinetics studies were not applied to other doses as presented here despite patch dosage was normalized to 1.6-fold that of oral dose in patients under active bisopronolol treatment. Please discuss, and further expand on the limitations for not assessing serum drug concentrations in the current study.

→Thank you for your valuable comments. Reference 5 does not describe the AUC, but we presented unpublished data. As you pointed out, the lack of measurements of blood drug concentration is a limitation of our study; we have added it to the text on limitations.

  • Methods line 47 - Your definition of chronic heart failure sounds more like an inclusion criteria rather than a definition of "chronic heart failure". For trials, please clearly identify your inclusion/exclusion criteria i.e. Chronic heart failure patients currently on oral bisopronolol treatment were recruited. This study did not exclude patients on other medications (diuretics,... etc.) to treat their heart failure conditions. 

Also, it isn't clear if there was a clear definition on how long patients had to be on oral bisopronolol before they could be transited to patch mode of administration. Was the limiting factor as to how long patients needed to be on this dosage before enrolment in the trial established? 

→Thank you for your valuable comments. Rather than defining chronic heart failure, we considered it more important to clarify the study population of this study, but we have revised the descriptions.

  • Figure 1 - Patients were excluded for adverse effects of the patch. However, pg 3 line 81 also states other following adverse effects were recorded, and discontinuation of the patch was decided by the attending physician (which supposedly resulted in the withdrawal of 4 patients with skin rash and 1 patient with bradycardia). What were the proportions of adverse effects reported (if any at all) after transiting to patch that were supposedly reported (other than 4x skin rash and 1x bradycardia)?

→We apologize for the confusing descriptions. All adverse effects occurred after patients switched from oral medication to patches. We deleted the group name in front of the onset of adverse effects in Figure 1.

  • Statistics - There is not enough information to evaluate suitability of statistics. Firstly, this study utilizes a paired study cohort (given that all 50 patients were prior oral bisopronolol users before transition onto dermal patches). In order to perform parametric analyses, authors should check if dataset conforms to normal distribution, and test for equal variance (Welch's correction applies to unequal variances).

What was the post-hoc analyses to test multiple comparisons in ANOVA? For student t-test, was this a one- or two-tailed, paired or unpaired t-test?

→Thank you for your valuable comment. We have re-analyzed the data by using suitable tests.

  • Were there differences in any findings between the 3 doses of bisopronolol (2 vs. 4 vs. 8mg)? I note that the dataset was probably underpowered to assess such a breakdown (which is a potential limitation). It will be curious to know if severity depended on the dosage of therapy.

→Thank you for this important comment. However, because the number of patients in this study was small, it is difficult to evaluate any differences between doses. We have included this point as a limitation.

6) Figure 2 - (i) Y-axis should be labeled as Heart rate (bpm) and bpm should be explained as beats per minute in the legend. (ii) Bar graphs do not do justice to individual data points. Suggest to presenting as box-violin plots which provides inter-subject variability information. (iii) colour scheme should be revised. It is hard to tell the error bars of the patch group against the blue shades background. 

Same for data presentation in Figure 3. 

→We thank you for your useful advice. There are many non-parametric results in this study. To improve readability, we now present the data from Figure 2 as a table (new Table 3). Furthermore, we have modified Figure 3 to present the data as a graph containing the median and 25th and 75 percentiles obtained by the Wilcoxon signed rank test.

  • Table 3 - Tablet vs patch should be highlighted at the top to indicate that Tablet group was conducted at baseline (0mo) whereas Patch was conducted over both (3 and 6mo) for clarity which phase of the treatment that these measurements were conducted. Similarly, all abbreviations including E/e' should be explained in footnote.

→Thank you for your advice. We have revised the manuscript per your instructions.

  • Discussion - Structure of discussion needs to be improved upon. It started off with saying patch group has <HR and <PVC which are findings from this current study. This is followed by results of blood concentration levels of bisoprolol (which was not conducted in the current study)...  cited reference  (ref #6) utilized 2.5mg tablet vs. 4mg patch. How does this actually translate to the other dosages engaged in the current study?

→Thank you for your useful comment. This study did not measure blood concentrations, so we cannot clearly demonstrate the efficacy of the drug. This is one of limitations of this study, and we now mention it as such.

Line 186-188: Sentence structure needs revision. Appears to suggest that the mechanism of bisoprolol (beta-blocker) increased cardiac contractions and heart rates, which is a contradiction.

→We apologize for the incorrect description and have revised the manuscript accordingly.

Line 193-194: Revise sentence - I don't understand the relevance of "Ten" 10 patients receiving mexiletine hydochloride and the implication of the subsequent sentence of reducing anti-arrhythmic drug dose and decrease in sudden death in context of your discussion?

→We apologize for the ambiguous descriptions. Ten patients received mexiletine hydrochloride for the treatment of PVC. We have clarified that those patients stayed on the same regimen and had no dose reduction or increase.

  • Discussion line 198 to 201 - Surprisingly, new results are presented in the discussion. This result is weird... and sentence structure should be amended to better communicate the findings. 

39 patients used rubber adhesives vs 26 patients used acrylic adhesives = 61 patients. Obviously, study recruitment was 50, so some alternated in between (n=11). However subsequent sentence state 11 patients used only acrylic adhesive, but 15 switched from rubber to acrylic. What about patients that switched from acrylic to rubber? and why am i having to calculate the proportions when this should be clearly presented i.e. how many used rubber adhesive throughout/ acrylic adhesive throughout/ alternated from rubber>acrylic adhesive/ acrylic>rubber adhesive? was there any control/regulation as how these were decided i.e. were patients randomly given a mix of the patches (acrylic/rubber) or did patients have side effects and therefore required to change patches?

→We apologize for the confusing descriptions. We also noticed calculation error(s). We have carefully revised and corrected the manuscript.

Minor comments:

  • Abstract line 16 - consider changing text to 06:00, 12:00, 18:00, and 00:00 hours or AM/PM instead of "o'clock".

→Thank you for your valuable advice. We have corrected the manuscript accordingly.

2) Table 1. All abbreviations should equally be explained in the footnote, rather than just expanding "ACE".

→Thank you for your comment. We have listed all abbreviations in the footnote section.

Round 2

Reviewer 1 Report

After the corrections have been made, the manuscript merits publication.

Author Response

Thank you for your important opinion

Reviewer 3 Report

The authors' have answered my comments to the best of their ability.

As mentioned, a significant limiting factor of this study was the difference in loading dose of bisoprolol where the patches were loaded with 1.6-fold that of the oral tablet dose. This was in the absence of any pharmacokinetics with the dose/mode of administration. Henceforth, it is unclear if the observation of reduced heart rate and/or reduction in atrial/ventricular ectopy is due to the increased doses or the extended bioavailability with the transdermal mode of administration (or both). It is hard to conclude that transdermal is therefore better than oral.

Minor comment:

1) Abstract - I'm not sure arrhythmia is the best description for a study that only studied ectopy counts (even though ectopic rhythm is considered an arrhythmia - but generally benign), as I'd be expecting counts of AF/AT, bigeminy/trigeminy and any other rhythmic diagnosis (although these are generally better suited for longer recording durations perhaps). Were these assessed? If not, please consider changing "arrhythmia" to "ectopic rhythm".

2) Table 2 - Under "Heart rate (bpm)" subheading - not sure "total" is a suitable sub sub heading, as that will imply "total heart rate". If I'm understanding this correctly, it should be called "total heart beats" in the 24h period and units should be stated as "(n)"?

3) Table 3 heading - consider saying "... maximum hourly heart rate (bpm) in 6-hourly blocks at 00:00, 06:00, 12:00, and 18:00."  

Author Response

Dear Reviewer 3

The authors' have answered my comments to the best of their ability.

As mentioned, a significant limiting factor of this study was the difference in loading dose of bisoprolol where the patches were loaded with 1.6-fold that of the oral tablet dose. This was in the absence of any pharmacokinetics with the dose/mode of administration. Henceforth, it is unclear if the observation of reduced heart rate and/or reduction in atrial/ventricular ectopy is due to the increased doses or the extended bioavailability with the transdermal mode of administration (or both). It is hard to conclude that transdermal is therefore better than oral.

→Your comment is quite important. We included this point in Discussions.

Minor comment:

1) Abstract - I'm not sure arrhythmia is the best description for a study that only studied ectopy counts (even though ectopic rhythm is considered an arrhythmia - but generally benign), as I'd be expecting counts of AF/AT, bigeminy/trigeminy and any other rhythmic diagnosis (although these are generally better suited for longer recording durations perhaps). Were these assessed? If not, please consider changing "arrhythmia" to "ectopic rhythm".

→We thank you for very important comment. As you indicated, there was no arrythmia requiring treatment such as AF/AT nor bigeminy/trigeminy. However, we consider an ectopic event as an arrhythmia. We changed arrythmias to PVC in Abstract.

2) Table 2 - Under "Heart rate (bpm)" subheading - not sure "total" is a suitable sub sub heading, as that will imply "total heart rate". If I'm understanding this correctly, it should be called "total heart beats" in the 24h period and units should be stated as "(n)"?

→Your comment is correct. We thank you very much. We made the revision accordingly.

3) Table 3 heading - consider saying "... maximum hourly heart rate (bpm) in 6-hourly blocks at 00:00, 06:00, 12:00, and 18:00."

→We revised the manuscript as you pointed out. Thank you very much.
